# Ultrasonic Phased Array Imaging Approach Using Omni-Directional Velocity Correction for Quantitative Evaluation of Delamination in Composite Structure

**DOI:** 10.3390/s23041777

**Published:** 2023-02-04

**Authors:** Xiangting Xu, Zhichao Fan, Xuedong Chen, Jingwei Cheng, Yangguang Bu

**Affiliations:** 1Institute of Process Equipment and Control Engineering, College of Mechanical Engineering, Zhejiang University of Technology, Hangzhou 310014, China; 2National Safety Engineering Technology Research Center for Pressure Vessels and Pipeline, Hefei General Machinery Research Institute Co., Ltd., Hefei 230031, China

**Keywords:** phased array, velocity correction, genetic algorithm, full matrix capture, CFRP

## Abstract

The ultrasonic detectability of buried defects within composite materials is dependent on the anisotropy of the composite material by which the propagation property of acoustic wave in each direction is variably affected. In this study, the characteristics of acoustic waves propagating in different directions for composite materials are explored based on the full matrix capture (FMC) data using an ultrasonic phased array. The elastic constant of multidirectional carbon fiber reinforced plastic (CFRP) laminate is first derived based on the genetic algorithm. The characteristics of transmitted and reflected waves in higher angles are predicted by implementing the Christoffel equation, and the focal law used in post-processing of FMC data can be optimized accordingly. The imaging results of the total focusing method (TFM) using the improved focal law are compared with the results of the conventional TFM. The results suggest that the optimized TFM can effectively characterize the defect by reducing the background noise. Furthermore, since it is impractical to theoretically correct angle-dependent velocity for in situ inspection, a linear extrapolation method based on the experimentally measurable velocity at low angles is proposed to estimate the velocity profile at higher angles. The imaging results using the fast extrapolated velocity profile is then compared with the theoretical, and it has been demonstrated that while the difference between the images using the theoretical focal law and the linearly extrapolated one is barely visible, the later one is overwhelmingly advantageous to be realiszd for engineering practices.

## 1. Introduction

Carbon fiber reinforced plastic (CFRP), in which the lightweight, high strength, and preferable toughness are provided [1,2], has been broadly used in aircraft and hydrogen fuel cell vehicles (e.g., airfoils and composite pressure vessels). However, delamination within CFRP structures arising from complicated manufacturing processes and extreme loading conditions may occur before and after service, leading to a reduction in the mechanical properties of the materials and a threat to the structural integrity [3]. Therefore, the effective non-destructive evaluation (NDE) of delamination is necessary for the avoidance of catastrophic accidents for those critical engineering structures [4].

Ultrasonic phased array testing is one of the most promising NDE approaches for sizing and locating buried delamination defects from composite structures. Among the imaging approaches recognized by industries, the total focusing method (TFM) thoroughly exploits full matrix capture (FMC) data from each transmit-receive pair of array elements to visualize the defects in an image [5]. However, for CFRP laminate, the anisotropy existing in the composite materials gives rise to complex wave propagating behaviors [6,7], so that the wave speed corresponding to each propagation path between array elements and target scatterer should be inconstant, and consequently, the artifacts and vague indication of defects are frequently seen in the ultrasonic images.

For CFRP laminates in which inhomogeneous materials are stacked in sequence, it is difficult to determine the focal law of the phased array due to the considerable change in velocity at different transmission and reflection angles. Recent advances in phased array techniques are predominantly seen in different ray tracing methods. Zhou et al. [8] optimized the algorithm parameters by combining the Dijkstra algorithm with Snell’s law to improve the efficiency and accuracy of the computation and experimentally estimated the wave propagation path in multi-layer materials. Li et al. [9] predicted the wave propagation path in CFRP laminate by using the Dijkstra algorithm, followed by the detection of a side-drilled hole with a diameter of 1.5 mm. Some previous works studied the frequency-wavenumber domain imaging method [10], implemented a three-dimensional Fourier transform on FMC data, and imaged the laminated structure of uniform materials by developing the phase shift migration method [11] and the interpolation algorithm in the wavenumber domain [12]. Although those methods are faster in frame rate relative to the TFM algorithm, they are not applicable to inhomogeneous CFRP materials. Mohammadkhani et al. [13] utilized wavelet transform for reducing coherent noise arising from multiple echoes from boundaries of CFRP layers, which improved the detectability of defects.

As a consequence of the complexity of the velocity profile for multidirectional CFRP laminate [14], prior studies generally simplified it to unidirectional CFRP material [15]. In order to measure the change in angle-dependent velocity of unidirectional CFRP material with increasing propagation angle, some measurement methods with single-sided access have been proposed as follows. The Backwall Reflection Method (BRM) [16] can measure the group velocity profile at a small range of angles by identifying the arrival time of the backwall signal transmitted and received by each array-element pair. This method has been tested by industries for CFRP in situ inspection [17]. Grager et al. [18] obtained 48 group velocities in a different direction through BRM calculations, and the maximum angle was found to be 58°. Then, the cubic polynomial function was used to fit data points within a small range of angles, and the signal-to-noise ratio of TFM images using the corrected delay law was improved. Cao et al. [19] determined the ray path and travel time in the corner part of an L-shaped CFRP laminate by using the quadric polynomial function of wave propagation angle and Dijkstra’s algorithm and eventually optimized the image in terms of signal-to-noise ratio. Some studies found that the elastic constants of unidirectional CFRP determined by particle swarm optimization algorithm using group velocity profile measured through BRM exhibited little difference from the actual ones obtained from the tensile test [20]. In addition, a forward numerical method in which the calculations of the group velocity profile of CFRP laminate necessitates the measurement of multiple physical parameters including anisotropic stiffness matrix, density, and lay-up direction was proposed by using the commercially available module of COMSOL Multiphysics [21].

The effective ultrasonic imaging of delamination defects within anisotropic CFRP materials necessitates the in situ measurement of omni-directional velocity (i.e., velocity profile in a wide range of angles) without complicated bespoke setups and time-consuming computations. In the present paper, the BRM method is first implemented to output the group velocity profile at low angles, followed by extrapolation of those at high angles using the Christoffel equation in conjunction with the genetic algorithm, thereby optimizing the focal law used in TFM imaging. Finally, the imaging efficacy on localization and quantification of CFRP delamination is examined systematically by the analysis of spatial amplitude distributions from four types of velocity profiles (termed constant velocity, piecewise constant velocity, linear velocity, and theoretical velocity respectively).

## 2. Theory and Method

### 2.1. BRM (Low-Angle Velocity Measurement)

For an ultrasonic transducer with array pitch size, *p,* and the total number of array elements, *M*, it is assumed that the transmitting element is *T*, the receiving element is *R*, and the number of time points is *n*. Broadly, the classical full matrix data is a three-dimensional matrix of $M_{T}\times M_{R}\times n$. In order to realize the measurement of low-angle velocity and minimize the contributions of the abnormal data, the FMC data as shown in Figure 1a is transformed into a three-dimensional matrix of $M_{T}\times M_{\Delta }\times n$, where the difference between the *T* and *R* is defined as ∆ = (|R−T|). The average of the two signals is taken in case of $\left|R-T_{i}\right|=\left|R-T_{j}\right|$. The time traces of different transmitting elements and the equivalent ∆ are accumulated, and the average is taken to obtain the two-dimensional matrix of $M_{\Delta }\times n$ as illustrated in Figure 1b.

As a consequence of the directional-dependent attenuation effect in heterogeneous CFRP materials, the waveform of the defect signal with the smaller ∆ is clearly distinguished from that of noise, while the defect signals of higher ∆ are often hidden in the background noise, delivering an inability to the recognition and extraction of defect signals (i.e., the time that signals scattered from defects arrive at the receiving element becomes unmeasurable). Therefore, the defect information from time traces of high ∆ is not considered here (see Figure 2).

According to the BRM method, a single array transducer is directly contacted on a flat CFRP laminate in which the front and back walls are parallel for both transmission and reception. In the Christoffel equation, the quasi-*P* wave velocity of unidirectional layup is symmetric along the normal direction of the CFRP laminate surface [20]. Hence, assuming that the transmitting elements and receiving elements are located at $\left(x_{i},0\right)$ and $\left(x_{j},0\right)$, respectively, the backwall reflection should occur at the position $\left(\left(x_{i}+x_{j},0\right),h\right)$, where *h* is the thickness of CFRP laminate. Given that the array element spacing is Δ×p, the wave speed can be expressed as
(1)$$v\left(\Delta \right)=\frac{2\sqrt{h^{2}+{\left(\frac{\Delta \times p}{2}\right)}^{2}}}{t_{\Delta }}$$
where $t_{\Delta }$ represents the time of flight from the back wall to the receiving element for the spacing ∆.

The ∆ is then substituted with the angle of back wall reflection *θ*, which is given mathematically by:(2)$$\Delta =\frac{2h\times tan\;\theta }{p}$$
(3)$$\theta =\arctan\left(\frac{\Delta \times p}{2h}\right)$$

By combing Formula (2) with Formula (3), the wave speed with respect to the angle *θ* may be concisely written as:(4)$$v\left(\theta \right)=\frac{2h}{t_{\theta }cos\theta }$$

### 2.2. High-Angle Velocity Measurement

CFRP materials are often thought to be isotropic along the fiber direction, such that the wave propagating speed should be independent of the direction in this plane. The elastic stiffness matrix *C* can be expressed as follows:(5)C=[C11C12C13C12C11C13C13C13C33  C44   C44   C66]Gpa
where $C_{12}$ = ($\;C_{11}-2C_{66}$) is the dependent component of elastic stiffness matrix, and the five independent elastic constants are $C_{11},\;C_{13},\;C_{33},\;C_{44}$ and $C_{66}$, respectively.

The relationship between the wave velocity profile of unidirectional CFRP layup whose elastic stiffness matrix can be calculated using the Christoffel equation [22]:(6)$$(\rho v^{2}\delta_{ij}-C_{ijkl}n_{k}n_{l})u_{i}=0$$
where ρ is the material density, v is the propagating speed, $\delta_{ij}$ is the Kronecker function, $C_{ijkl}$ is the elastic stiffness matrix, $n_{k}$ and $n_{l}$. denote the wave propagating direction, and $u_{i}$ is the particle vibration direction. Under the assumption of $\Gamma_{ik}=C_{ijkl}n_{j}n_{l}$ (namely the Christoffel tensor), it can be written mathematically as:(7)$$\;\;\;\;\;\;\;\;\;\;\;\;\;\;\Gamma =\left[\begin{array}{ccc} C_{11}n_{1}^{2}+C_{66}n_{2}^{2}+C_{44}n_{3}^{2} & \left(C_{11}-C_{66}\right)n_{1}n_{2} & \left(C_{13}+C_{44}\right)n_{1}n_{3}\\ \left(C_{11}-C_{66}\right)n_{1}n_{2} & C_{66}n_{1}^{2}+C_{11}n_{2}^{2}+C_{44}n_{3}^{2} & \left(C_{13}+C_{44}\right)n_{2}n_{3}\\ \left(C_{13}+C_{44}\right)n_{1}n_{3} & \left(C_{13}+C_{44}\right)n_{2}n_{3} & C_{44}+\left(C_{33}-C_{44}\right)n_{3}^{2} \end{array}\right]$$

In the case of ultrasonic wave propagating along the direction *n* = [(cos φ,sin φ, 0)] in the isotropic plane of CFRP materials, the Christoffel tensor can be simplified as follows:(8)$$transmit–receive\;\;\;\;\;\;\;\;\;\;\;\;\;\;\;\;\;\;\;\;\;\;\;\;\;\;\;\;\;\;\;\;\;\;\;\;\;\;\;\;\;\Gamma =\left[\begin{array}{ccc} C_{11}cos^{2}\varphi +C_{66}sin^{2}\varphi & \left(C_{11}-C_{66}\right)cos\varphi sin\varphi & 0\\ \left(C_{11}-C_{66}\right)cos\varphi sin\varphi & C_{66}cos^{2}\varphi +C_{11}sin^{2}\varphi & 0\\ 0 & 0 & C_{44} \end{array}\right]$$

By combining the Christoffel tensor with Equation (8), the non-singular solution of the Christoffel function based on elastic constant and propagation direction is calculated accordingly as:(9)$$v_{L}=\sqrt{\frac{C_{11}}{\rho }}$$
(10)$$v_{T1}=\sqrt{\frac{C_{44}}{\rho }}$$
(11)$$v_{T2}=\sqrt{\frac{C_{66}}{\rho }}$$
where $v_{L}$ denotes longitudinal velocity and $v_{T1}$ and $v_{T2}$ are transverse velocities. Note that the two-dimensional plane is isotropic and wave velocity is merely dependent on the elastic constant and material density regardless of propagation direction.

When ultrasonic wave propagates along the direction *n* = [ (cosθ,0,sinθ)], the Christoffel tensor may be expressed below:(12)$$\Gamma =\left[\begin{array}{ccc} C_{11}cos^{2}\theta +C_{44}sin^{2}\theta & 0 & \left(C_{13}+C_{44}\right)cos\theta sin\theta \\ 0 & C_{66}cos^{2}\theta +C_{44}sin^{2}\theta & 0\\ \left(C_{13}+C_{44}\right)cos\theta sin\theta & 0 & C_{44}+\left(C_{33}-C_{44}\right)sin^{2}\theta \end{array}\right]\;$$

Subsequently, by calculating the three eigenvalues of v, the velocity of the quasi-longitudinal wave $v_{qP}$ is calculated by:(13)$$v_{qP}=\sqrt{\frac{M+\sqrt{M^{2}-4N}}{2\rho }}$$

Likewise, the velocities $v_{qSH}$ and $v_{qSV}$ of the two types of quasi-shear waves are computed as follows:(14)$$v_{qSH}=\sqrt{\frac{C_{44}sin^{2}\theta +C_{66}cos^{2}\theta }{\rho }}$$
(15)$$v_{qSV}=\sqrt{\frac{M-\sqrt{M^{2}-4N}}{2\rho }}$$
where *M* and *N* are defined by Equations (16) and (14):(16)$$N=C_{11}cos^{2}\theta +C_{33}sin^{2}\theta +C_{44}$$
(17)$$N=(C_{11}cos^{2}\theta +C_{44}sin^{2}\theta )(C_{44}cos^{2}\theta +C_{33}sin^{2}\theta )-{\left(C_{11}+C_{44}\right)}^{2}sin^{2}\theta cos^{2}\theta$$

For unidirectional plies in a repeating sequence of two orientation angles (e.g., 0°and 90°), when the alignment of one-dimensional array elements is parallel to the plies with a fiber orientation of 0°, the 0° plies are thought to be elastically anisotropic, and the 90° ones, of which the wave propagation is seen on the cross-section, may be approximate to isotropic. According to Snell’s law, the relationship between their velocities can be expressed as:(18)$$\frac{\sin\left(\theta_{1}\right)}{\sin\left(\theta_{2}\right)}=\frac{v_{1}}{v_{2}}=\frac{f_{1}\left(C,\theta_{1}\right)}{f_{2}\left(C\right)}$$
where $v_{1}$ is wave velocity of 0° plies as a function of elastic stiffness matrix C and *θ*_1_ described in Equation (13) and $v_{2}$ is wave velocity of 90° plies as a function of elastic stiffness matrix. Note that $v_{2}$ is independent of *C* under the assumption of homogeneity. Therefore, Equations (16) and (17) can be simplified to $M=C_{11}+C_{44}$ and $N=C_{11}C_{44}$, respectively. Accordingly, the velocity $v_{qP}$ = ($\sqrt{\frac{C_{11}}{\rho }}$) is obtained from Equation (9), implying that wave velocity on the CFRP isotropic plane indicated by Equation (6) is identical to that on the anisotropic plane in the direction *θ* of 0°.

In practice, it is difficult to measure the refraction angles of the plies with different directions. In order to facilitate the actual measurement, *θ*_0_ is employed to represent the angle between the line from the incidence to the back wall reflection and the normal to the back wall. Specifically, *θ*_0_ may be defined by:(19)$$tan\theta_{0}=\frac{h_{1}*tan\theta_{1}+h_{2}*tan\theta_{2}}{h_{0}}$$
where *θ*_1_ and *θ*_2_ are the refraction angles of 0° and 90° plies, respectively. In addition, *h*_1_ and *h*_2_ are the thickness of 0° and 90° plies, respectively.

Since *θ*_0_ can be calculated by the arc tangent of the ratio between the element spacing and the thickness of the test block *h*_0_, the refraction angles *θ*_1_ and *θ*_2_ in the plies of different directions can be obtained from Equations (18) and (19) if the *C*, *h*_1_ and *h*_2_ are known. Subsequently, the change in wave velocity with increasing angle in the 0° plies may be estimated, whereas it is impractical to express *θ*_1_ and *θ*_2_ in terms of a formula containing the elastic constant *C* through Equations (18) and (19). Consequently, *θ*_2_ can be written as a function of *θ*_1_:(20)$$\theta_{2}=\arctan\left(\frac{h_{0}*tan\theta_{0}-h_{1}*tan\theta_{1}}{h_{2}}\right)$$

Since the time-of-flight *t_exp_* at *N* different angles has been experimentally measured, it is compared with the theoretical one *t_theo_* defined in Equation (17) to determine the optimal refraction angle *θ*_1_, which is thought to be the actual refraction angle. Under the assumption that the 90° ply is isotropic, the relation curve between velocity and angle for the 0° ply can be computed by Equations (18) and (20).
(21)$$2*\left(\frac{h_{1}}{cos\left(\theta_{1}\right)*v_{1}}+\frac{h_{2}}{cos\left(\theta_{2}\right)*v_{2}}\right)=t_{theo}$$

When the elastic constant is provided, the relationship between wave velocity and angle can be described by Equation (13). Alternatively, five independent elastic constants may be determined from the magnitude of velocity at different angles. However, as a consequence of the nonlinear relationship between velocity and elastic constant, the classical function cannot be solved explicitly, so that it necessitates an approach optimized by a nonlinear algorithm. The genetic algorithm (GA) is an adaptive heuristic search algorithm based on natural selection theory, which delivers effectiveness on solving such global optimization problems. In this study, the genetic algorithm is implemented to particularly solve the problems of low precision and premature convergence to local optima in the inversion analysis. The least squares method is usually used to construct the loss function *F*, so as to minimize the error between the experimental value and the theoretical value. Thus, the optimal solution of the elastic constants can be provided by *F*, which is defined by:(22)$$F=\overset{M}{\underset{i=1}{\sum }}{\left(v_{1}^{exp}\left(\theta_{i}\right)-v_{1}^{theo}\left(\theta_{i}\right)\right)}^{2}$$
where $v_{1}^{exp}\left(\theta_{i}\right)$ is the wave velocity in a function of angle for 0° ply which is output by the experimentally measured equivalent velocity through Equations (18)–(21). $v_{1}^{theo}\left(\theta_{i}\right)$ is the theoretical value calculated from Equation (13) according to the Christoffel equation, and *M* is the number of selected angles.

In wave velocity $v_{1}^{’}$ at large angles of 0°, ply is predicted by the Christoffel equation. Since velocity is in close proximity to constant for a 90° ply, the equivalent velocity $v_{0}^{’}$ can be written as:(23)$$v_{0}^{’}=\frac{\frac{h_{0}}{cos\left(\theta_{0}\right)}}{\frac{h_{1}}{cos\left(\theta_{1}\right)*v_{1}^{’}}+\frac{h_{2}}{cos\left(\theta_{2}\right)*v_{2}}}$$

The flowchart for the optimization process is shown in Figure 3.

TFM is known as one of the most effective ultrasonic imaging methods widely used in non-destructive testing [23]. First, the target imaging area is discretized into a mesh grid [24], and the time-of-flight between the target location and each array element is calculated (see Figure 4). Subsequently, the signal amplitude of interest can be estimated by interpolating the FMC data with the computed time-of-flight. Therefore, the imaging result at every single pixel are output by superimposing all the amplitudes from each transmit-receive pairs [25]. The calculations for the intensities of TFM image can be expressed by Equations (24) and (25) [26]:(24)$$t_{ij}=\frac{\sqrt{{\left(x_{Ti}-x\right)}^{2}+{\left(y_{Ti}-y\right)}^{2}}}{c}+\frac{\sqrt{{\left(x_{Rj}-x\right)}^{2}+{\left(y_{Rj}-y\right)}^{2}}}{c}$$
(25)$$I_{ij}=\overset{n}{\underset{i=1}{\sum }}\overset{n}{\underset{j=1}{\sum }}S_{ij}\left(t_{ij}\right)$$
where *x* and *y* are the coordinates of target imaging points; $x_{Ti}$ and $y_{Ti}$ are the coordinates of the transmitter element, $x_{Rj}$ and $y_{Rj}$ are the coordinates of the receiver element, $t_{ij}$ is the time-of-flight between each pixel and the array elements, and *S* is FMC data.

## 3. Experimental Setup

The ultrasonic phased array probe (Imasonic, Voray-sur-l’Ognon, France) with 64 array elements, center frequency of 5 MHz (−6 dB bandwidth is 86% of the center frequency), and pitch size of 0.6 mm, as well as a commercial array controller (Peak NDT LTPA, Derby, UK) are used in the following study. A single probe is in direct contact with the test block covered by small amount of gel couplant to conduct pulse-echo measurements.

A 10 mm thick CFRP test specimen in which the nominal density was measured by the manufacturer as 1512 kg/m^3^ was designed and fabricated to contain the artificial defects with specified size. More specifically, the specimen contains 23 unidirectional plies of the same thickness stacked in a repeating sequence of two orientation angles (e.g., 0°and 90°). As shown in Figure 5, the release films were embedded during manufacturing to simulate the delamination defects, which were located at depths of 3 mm and 5 mm and had a diameter of 8 mm. Figure 6 displays the experimental setup of the phased array detection and cross-section of the bespoke CFRP test block.

## 4. Results and Discussion

First, the BRM was implemented to measure the time-of-flight of signals scattered from the delamination defect at different angles. Through experimental analysis, the reflections from the delamination were found to be measurable when the difference between the sequence number of the transmitter and the receiver ∆ was less than 28. A third-order polynomial was used to interpolate the measured points, and the fitting curve is shown in Figure 7.

According to Equations (14)–(17) and the time-of-flight displayed in Figure 7, discrete points of wave velocity for the specimen can be computed in case of an incident angle from 0° to 45.8° (see Figure 8a). Subsequently, the Christoffel equation in conjunction with the genetic algorithm are implemented to fit velocities at high angles. After 400 iterations performed, the best fitness and mean fitness value are found to be 104.654 and 104.656, respectively. Hence, the error is merely circa. 0.04%, which suggests that the fitting has realized an acceptable effect. Eventually, the elastic stiffness matrix for CFRP plies may be calculated by:C=[11.24C1213.18C1211.2413.1813.1813.1831.28  0.01   0.01   C66]GPa
where $C_{12}$=$\;C_{11}$ − 2$C_{66}$, and the calculations of velocity fitting is not affected by unknown $C_{12}$ ($C_{66}$) under the assumption of longitudinal wave measurement. The extrapolated results in which the wave velocities of the CFRP laminate at high angles are fitted by the genetic algorithm. It can be seen from Figure 8b that the wave velocities change gradually in the vicinity of 0° and rise sharply with increasing angles.

Next, based on above mentioned velocities at low angles, four different types of high-angle velocity prediction methods are proposed for a comparative study. As shown in Figure 9, the constant velocity (C) refers to the wave velocity at vertical incidence (i.e., angle of 0°) applied consistently to each angle, and piecewise constant velocity (P) denotes the constant wave velocity at high angles, equal to the maximum velocity at low angles, such that the curve at different angles exhibits good continuity. Furthermore, linear velocity (L) indicates that the high-angle velocities change linearly according to the first derivative at the critical point. Theoretical velocity (T) is the predicted high-angle velocities calculated from Equation (19). Note that all the non-discrete points of the curves are obtained by linear interpolation. While the low-angle velocities of the last three types of curves are the same, the high-angle velocities of the P curve is significantly different from those of other two curves, and the difference between T curve and L curve is less measurable.

Prior to TFM imaging, the signal was post-processed with a Gaussian filter (center frequency 5 MHz, −6 dB bandwidth). In the following section, two embedded delamination defects (replaced by release films) of 8 mm diameter at depths of 3 mm and 5 mm were imaged using TFM in conjunction with different optimized focal law. Their imaging results are displayed in Figure 10a–h. Since the existence of anisotropy (both at ply level and the fiber-matrix level) gives rise to backscattering or structural noise, the conventional TFM imaging features of delamination defects at a depth of 3 mm are inevitably masked by those near-field background levels in Figure 10a. Although the features at a depth of 6 mm can be clearly visualized by the second echoes of the 3 mm deep defect, there is a misinterpretation of spatial information of the defects. In addition, a vague indication of the back wall is presented in Figure 10a from which is can be seen that its backscattering is weak and discontinuous. The inability to identify the back wall and the near field (i.e., imaging area of depths less than 3 mm) target is also seen in Figure 10e, which suggests that the standard TFM using single velocity is not effective on heterogenous materials.

Despite the fact that the defects at depths of 3 mm and 5 mm can be seen in Figure 10b,f, respectively, the imaging performance of the lateral area is suppressed by constant velocities used for those high angles (i.e., either the shallower defect or the target below the lateral array elements might be undetectable). Furthermore, Figure 10c,d demonstrates that the optimized TFM using linear velocity and theoretical velocity can deliver overwhelmingly advantageous imaging efficacy for both the geometric features (e.g., back wall) and the 3 mm deep defect. Similarly, Figure 10g,h also exhibits accurate localization and sizing of 5 mm deep defects. Figure 10c,d as well as Figure 10g,h show barely visible differences between each of them, which might be attributed to specular reflections [27,28] of planar defects rather than point scattering [29,30] (e.g., pore-like defects) allowing omni-directional waves to be measured by different receiver elements.

In order to investigate the efficacy of TFM imaging using four different focal laws, the region of interest (i.e., in vicinity of the delamination defects) is extracted for conducting in-depth analysis of localization and sizing. Figure 11a,c displays the intensities of two delamination defects at depths of 3 mm and 5 mm, respectively, along *z*-axis at *x* = −1.1 mm, thereby indicating the depth of defects. It is worth noting that the intensities using four different velocity profiles (C, P, L, and T) are in close proximity to each other, delivering accurate position of the defects. However, the ones with velocity profile C exhibit inconsistent changes in the near field (i.e., *z* is less than 5 mm) due to more contributions from background noise. The negligible difference between the ones using P, L, and T profiles is thought to arise from significant energy absorption of the CFRP specimen giving rise to minimum involvement of high-angle scattered waves.

Furthermore, Figure 11b,d, present the sizing analysis by observing intensities along *x*-axis at *z* = 3 mm and *z* = 5 mm. Apart from the results from the velocity profile C, the quantification of the defect using the P, L, and T profiles velocities is seen to deliver appropriate consistency with the actual defect size. From Table 1 and Table 2, in the case of known size of the defect, the average intensity of the defect can be calculated by taking the mean value of the pixels belonging to the actual position of the defects. More importantly, the average intensity with T is seen to be approx. 9 dB and 7 dB higher than that with C for the 3 mm deep defect and 5 mm deep defect, respectively. Additionally, the differences between average intensity of P and L profiles as well as L and T profiles is at least 1 dB. In summary, the optimized TFM imaging using the velocity profiles of T and L undoubtedly prevail over the other two. For engineering practice, it might be inefficient to mathematically resolve the elastic constants using the Christoffel equation on a case-by-case basis. Therefore, it is more convenient to replace the elastic constant by simple linear function based on the knowledge of velocity profile characteristics in this study (i.e., a compromise between imaging quality and frame rate).

## 5. Conclusions

In this paper, the velocity profile of CFRP materials in a low angle range is first measured by combining FMC data and the BRM method. Based on the Christoffel equation, the relationship between the elastic stiffness matrix of CFRP 0° ply and the velocity profile is established. Four independent elastic constants of CFRP materials can be obtained by th genetic algorithm. Therefore, the propagation characteristics of ultrasonic wave at a high angle have been determined to optimize the focal law. By comparing the optimized TFM images with the conventional ones, the results reveal significant improvements in signal-to-noise ratio for defects and geometric features realized by the implementation of the proposed theoretical velocity. Since it is impractical to theoretically calibrate high-angle velocities during in situ inspection, this paper also proposes an approach by simply utilizing a linear function in which the continuity of the first derivative at the critical point can be delivered to describe velocity changes at high angles based on experimentally measured low-angle velocities. Furthermore, the TFM results demonstrate that the linear velocity yields little difference in imaging performance relative to the theoretical one, whereas it should be more widely applicable due to the little computation involved. In our experiments, as a consequence of multiple reflections from the 3 mm deep delamination defect, the existence of some artifacts should be expected. The TFM technique based on these omni-directional velocity correction, therefore, both improves imaging efficacy and reduces the reliance of prior knowledge making it a viable prospect for in situ testing of, for example, composite overwrapped pressure vessel.

## Figures and Tables

**Figure 1 sensors-23-01777-f001:**
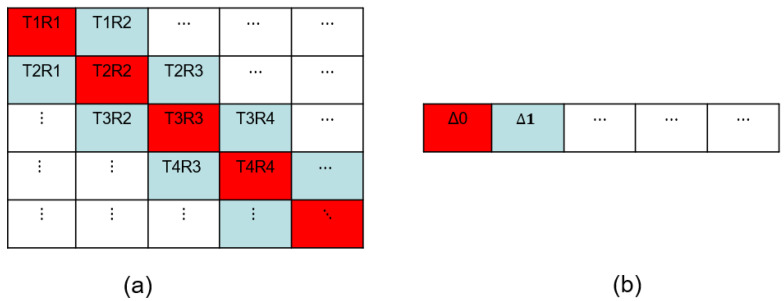
The diagram of (**a**) the three-dimensional matrix of $M_{T}\times M_{R}\times n$ and (**b**) the transformed two-dimensional matrix of $M_{\Delta }\times n$.

**Figure 2 sensors-23-01777-f002:**
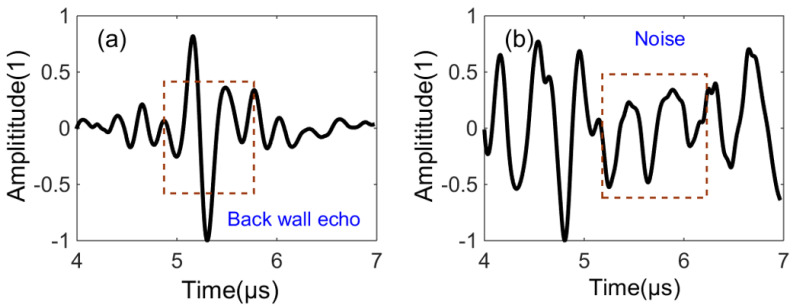
Normalized time domain waveform with different transmit-receiving intervals ∆, ∆ = 0 (**a**) and ∆ = 29 (**b**).

**Figure 3 sensors-23-01777-f003:**
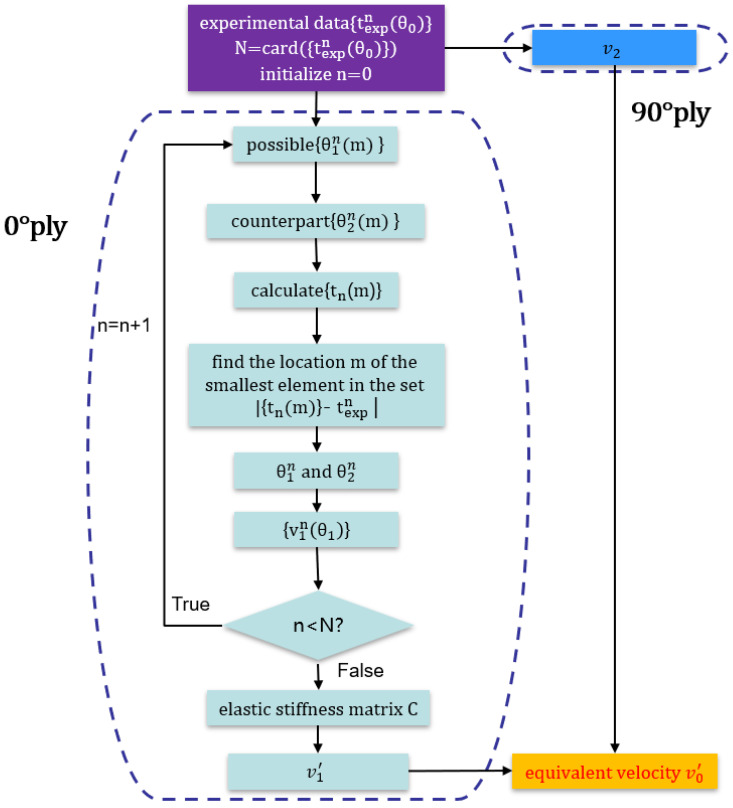
Flow chart of the velocity measurement for CFRP multidirectional plate.

**Figure 4 sensors-23-01777-f004:**
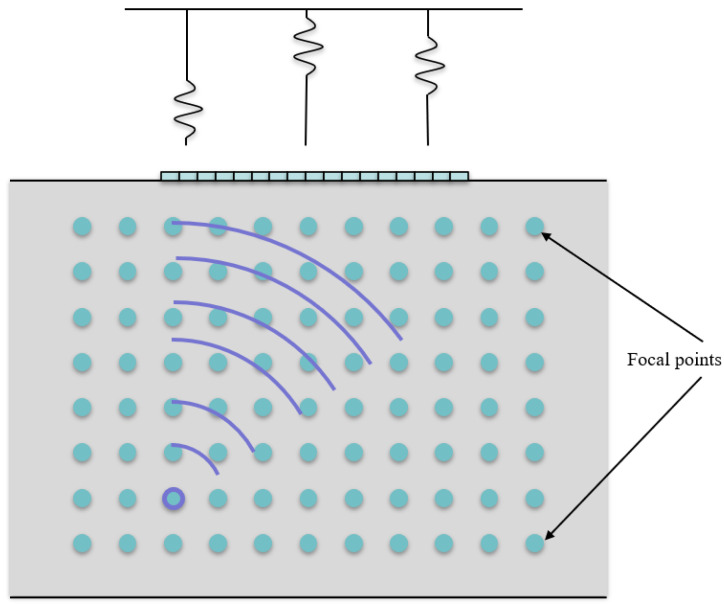
Schematic diagram of TFM imaging.

**Figure 5 sensors-23-01777-f005:**
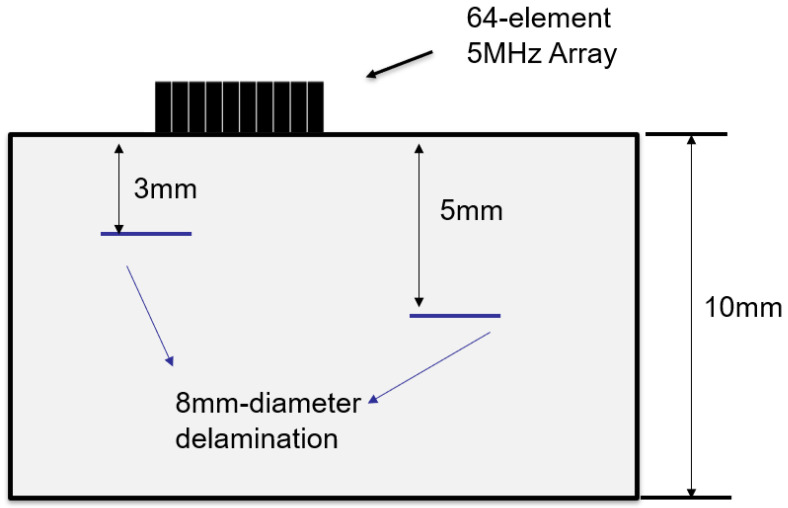
The diagram of the CFRP specimen containing delamination defects at different locations.

**Figure 6 sensors-23-01777-f006:**
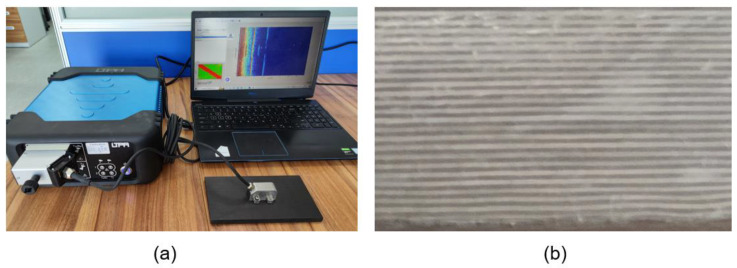
Experimental setup of the phased array detection (**a**) and a cross-section of the test block (**b**).

**Figure 7 sensors-23-01777-f007:**
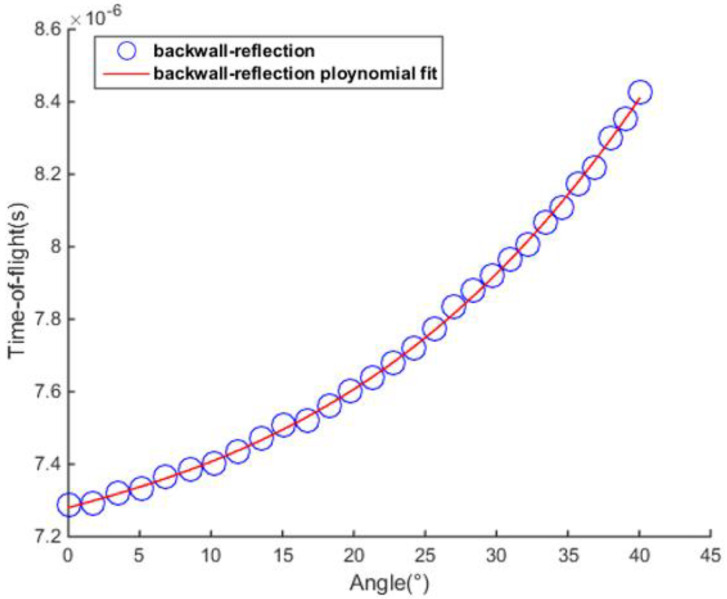
The cubic polynomial fitting curve from experimentally measured discrete points.

**Figure 8 sensors-23-01777-f008:**
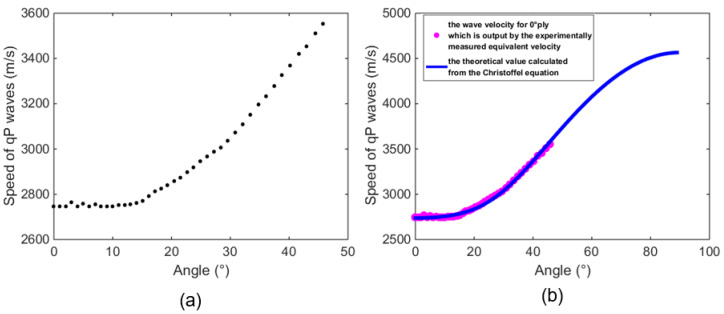
The diagrams of (**a**) measured discrete velocities at the angles from 0° to 45.8° and (**b**) the theoretical fitting curve from 0° to 90°.

**Figure 9 sensors-23-01777-f009:**
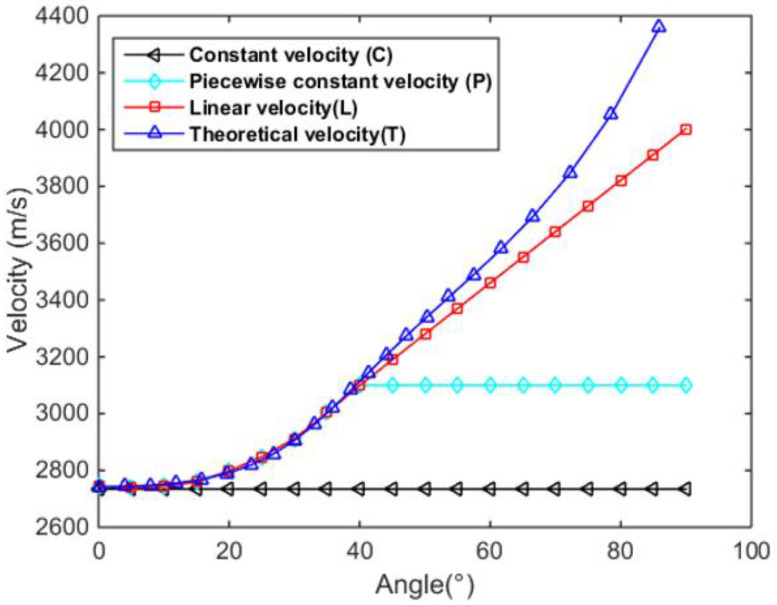
The predicted high-angle velocities using four different approaches.

**Figure 10 sensors-23-01777-f010:**
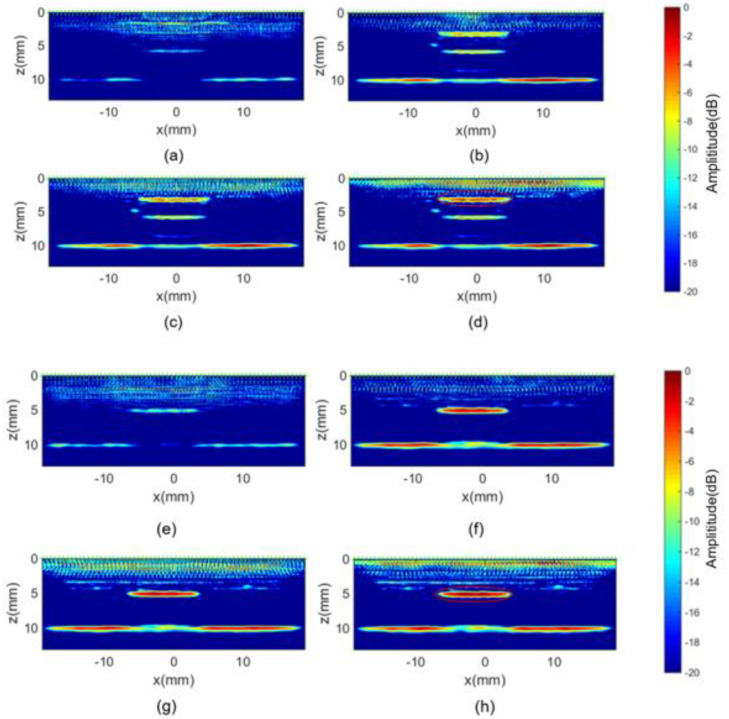
TFM imaging results for the 3 mm deep delamination defect using delay law of (**a**) constant velocity, (**b**) piecewise constant velocity, (**c**) linear velocity and (**d**) theoretical velocity as well as the 5 mm deep delamination defect using delay law of (**e**) constant velocity, (**f**) piecewise constant velocity, (**g**) linear velocity, and (**h**) theoretical velocity.

**Figure 11 sensors-23-01777-f011:**
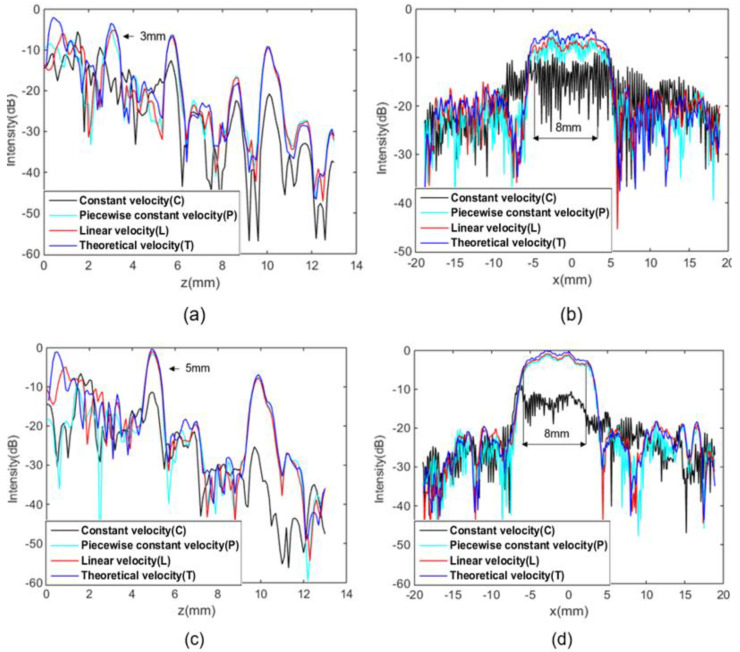
TFM imaging intensities for 3 mm deep defects along lines at (**a**) *x* = 1.1 mm and (**b**) *z* = 3 mm, as well as the 5 mm deep defects along lines at (**c**) *x* = 1.1 mm and (**d**) *z* = 5 mm.

**Table 1 sensors-23-01777-t001:** 3 mm deep defect intensities (dB).

	Constant Velocity (C)	Piecewise Constant Velocity (P)	Linear Velocity (L)	Linear Velocity (L)
Average intensity	−14.4254	−8.0159	−6.0752	−5.0412
Std of intensity	3.8881	1.5922	0.94358	0.86467
Average intensity	−20.4436	−22.1136	−20.7535	−21.0696
Std of intensity	5.5022	5.9473	5.6156	6.2298

**Table 2 sensors-23-01777-t002:** 5 mm deep defect intensities (dB).

	Constant Velocity (C)	Piecewise Constant Velocity (P)	Linear Velocity (L)	Linear Velocity (L)
Average intensity	−12.0681	−3.7233	−2.6446	−1.5202
Std of intensity	1.5661	0.8938	0.8842	0.9431
Average intensity	−23.4903	−25.8872	−24.3567	−23.473
Std of intensity	5.6004	7.1157	7.7815	7.4145

## Data Availability

Not applicable.

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
