# Peer review of "Ultrasonic Phased Array Imaging Approach Using Omni-Directional Velocity Correction for Quantitative Evaluation of Delamination in Composite Structure"

_sensors, 2023, doi:10.3390/s23041777_

Round 1
Reviewer 1 Report
Overall, the manuscript is in excellent condition and the results were presented well. I only have very minor suggestions. First, I would do a quick pass for spelling and grammar. There were few mistakes but a couple stood out. Second, I would keep graph labeling consistent (i.e. change Time/µs to Time (µs) on figure 2.) Third, I would make sure that all figures have units for every axis, even if the units are arbitrary (fig 2 and fig 10.) I would also bold most of the text on the graphs to make them more legible, especially the legends. I would also completely spell out experimental and theoretical in figure 8b. On figure 7, the graph should be consistent with the others and have borders on all sides, not just the bottom and left.
Reviewer 2 Report
The authors did a good job, the publication is at a high level. Research and results are valuable and important. The math tools are of a high standard.
The graphs in Figure 11 are hard to read.
Figure 4 caption cannot be bold.
